# The polarity protein Baz forms a platform for the centrosome orientation during asymmetric stem cell division in the Drosophila male germline

Mayu Inaba[1,2]*, Zsolt G Venkei[1], Yukiko M Yamashita[1,2]*

[1]Life Sciences Institute, University of Michigan, Ann Arbor, United States; [2]Department of Cell and Developmental Biology, School of Medicine, Howard Hughes Medical Institution, University of Michigan, Ann Arbor, United States

**Abstract** Many stem cells divide asymmetrically in order to balance self-renewal with differentiation. The essence of asymmetric cell division (ACD) is the polarization of cells and subsequent division, leading to unequal compartmentalization of cellular/extracellular components that confer distinct cell fates to daughter cells. Because precocious cell division before establishing cell polarity would lead to failure in ACD, these two processes must be tightly coupled; however, the underlying mechanism is poorly understood. In *Drosophila* male germline stem cells, ACD is prepared by stereotypical centrosome positioning. The centrosome orientation checkpoint (COC) further serves to ensure ACD by preventing mitosis upon centrosome misorientation. In this study, we show that Bazooka (Baz) provides a platform for the correct centrosome orientation and that Baz-centrosome association is the key event that is monitored by the COC. Our work provides a foundation for understanding how the correct cell polarity may be recognized by the cell to ensure productive ACD.

*For correspondence: minaba@umich.edu (MI); yukikomy@umich.edu (YMY)

## Introduction

Asymmetric division of adult stem cells that produces a self-renewing stem cell and a differentiating daughter cell is crucial for tissue homeostasis in diverse systems (*Morrison and Kimble, 2006*). Disruption of this balance is postulated to underlie many pathological conditions, including tumorigenesis/tissue hyperplasia (due to excess self-renewal) and tissue degeneration/aging (due to excess differentiation). Intensive investigation has revealed the mechanisms that polarize cells and orient the division plane; however, less is known about how cells might respond to perturbation of cell polarity and whether/how cells might ensure that cell division occurs only after the establishment of correct polarity.

A mechanism to coordinate the timing of two potentially independent events during the cell cycle is defined as a checkpoint. The spindle position checkpoint (SPOC) in budding yeast is a prominent example of a checkpoint that coordinates cell division with cell polarity; mitotic exit is delayed by the activity of SPOC if the spindles are not correctly oriented in a manner to ensure equal segregation of chromosomes into the mother and daughter cells (*Pereira and Yamashita, 2011*). A similar spindle orientation checkpoint mechanism is also reported in fission yeast (*Gachet et al., 2006*). Despite the importance of asymmetric divisions in the development of multicellular organisms, the potential checkpoint mechanisms that ensure asymmetric cell divisions, similar to the SPOC in the budding yeast, are poorly defined.

We have established *Drosophila* male germline stem cells (GSCs) as a model to study the checkpoint that coordinates polarization of cells and cell division (*Cheng et al., 2008*; *Inaba et al., 2010*; *Yuan et al., 2012*). *Drosophila* male GSCs divide asymmetrically, producing one stem cell and one differentiating cell, the gonialblast (GB). Asymmetric stem cell division is achieved by stereotypical positioning of the mother and daughter centrosomes in order to orient the spindle perpendicularly to

**eLife digest** The tissues of an animal's body are built from cells that are originally derived from stem cells. Each stem cell can divide and give rise to another stem cell and a cell that will become a more specific type of cell—such as a nerve cell, muscle cell, or sperm cell. If this asymmetric cell division is disrupted, it can result in developmental disorders and diseases such as cancer.

When a cell divides, a structure known as the spindle separates the copies of the chromosomes into the two newly formed cells. The spindle consists of long protein filaments that extend from two smaller structures known as centrosomes, which are found at opposite sides of the cell. The position of these centrosomes governs the orientation of the spindle, which in turn determines the plane in which cell division takes place. Thus, cells that need to divide with a certain orientation must have a mechanism that ensures that their centrosomes are correctly positioned. However, the existence of such a mechanism has been underexplored, and it remains unclear how the alignment of the centrosomes is controlled.

Inaba et al. analyzed how stem cells in the male fruit fly divide asymmetrically to form one stem cell and second cell that develops into sperm. The experiments revealed that a protein called Bazooka (or Baz for short) closely associates with the centrosomes just before the cells start to divide. Many other animals—such as humans and worms—have proteins that are closely related to Bazooka, which are needed for asymmetric cell divisions. When Inaba et al. reduced the levels of the Bazooka protein in the fruit fly cells, a large number of these cells ended up with centrosomes that were incorrectly aligned. As a result, these cells' spindles were also oriented incorrectly.

These findings suggest that the interactions between Bazooka and the centrosomes inform a cell when the centrosomes are correctly orientated. However, further work will be required to determine the details of how Bazooka controls asymmetric cell divisions.

the hub cells, the major component of the stem cell niche (*Yamashita et al., 2003*, *2007*). Stereotypical centrosome behavior that occurs in preparation for asymmetric cell division has been described in other stem cell systems (*Rebollo et al., 2007*; *Rusan and Peifer, 2007*; *Wang et al., 2009*; *Conduit and Raff, 2010*; *Januschke et al., 2011*; *Lu et al., 2012*; *Salzmann et al., 2014*), suggesting the evolutionarily conserved nature of the process. Asymmetric GSC division is further ensured by the centrosome orientation checkpoint (COC) that prevents mitotic entry in the presence of incorrectly oriented centrosomes (*Figure 1A*) (*Cheng et al., 2008*; *Inaba et al., 2010*; *Yuan et al., 2012*). Upon sensing the centrosome misorientation, COC is activated to prevent mitotic entry (*Figure 1A*). Thus, the defective COC can be suggested by the presence of misoriented spindles. We have shown that the centrosomal protein Cnn and a polarity kinase Par-1 are critical component of the COC, defects of which leading to high frequency of spindle misorientation (*Inaba et al., 2010*; *Yuan et al., 2012*).

The physical basis of correct centrosome orientation monitored by the COC remains a mystery. In the case of the spindle assembly checkpoint (SAC), the lack of microtubule attachment to the kinetochore (or tension at the kinetochore) is sensed as defective spindle assembly, triggering SAC activation to halt mitotic progression (*Musacchio and Salmon, 2007*). In the operation of the COC, what is sensed as correct or incorrect centrosome orientation to inactivate or activate the COC remains unknown. Here, we show that Bazooka (Baz)/Par-3, a well-established polarity protein and a known substrate of Par-1 kinase, forms a small subcellular structure that anchors the centrosome right before mitotic entry. We provide evidence that the association between Baz and the centrosome is the key event that is interpreted to indicate 'correct centrosome orientation' by GSCs. We further show that Par-1-dependent phosphorylation of Baz is critical for GSC spindle orientation. Our study provides a framework of the mechanism by which GSC sense correct cell polarity.

## Results

### Baz forms a subcellular structure between the hub and GSCs that closely associate with the centrosome

Baz/Par-3, which is a known physiological substrate of Par-1, contributes to cell polarity and spindle orientation in diverse systems (*Watts et al., 1996*; *Benton and St Johnston, 2003*; *Siller and Doe, 2009*).

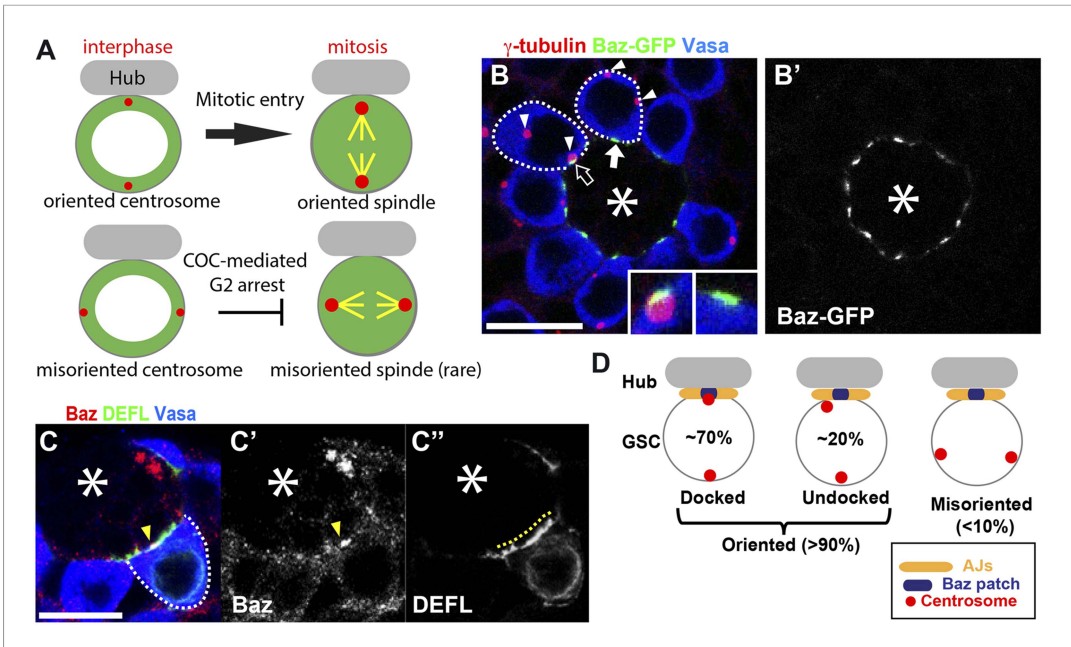

**Figure 1**. The apical centrosome associates with the Baz Patch. (**A**) The centrosome orientation in GSCs and the function of COC. (**B**) An example of an apical testis tip showing the Baz patch and centrosomes. The apical centrosome often associates with the Baz patch (open arrow). The Baz patch (solid arrow) remains in GSCs with misoriented centrosomes. Centrosomes are indicated with arrowheads. The insets show Baz patches with or without the centrosome. (**B'**) Baz-GFP only. Bar: 10 μm. The colored text indicates the fluorescence pseudocolor in the images in this and subsequent figures. The γ-tubulin staining indicates the centrosome. The Vasa staining indicates the germ cells. The hub is denoted with an asterisk. (**C**) The Baz patch is a small structure that is located on the GSC-hub interface. The arrowhead in (**C**, **C'**) indicates the Baz patch stained with anti-Baz (red). The yellow dotted line in (**C''**) indicates the GSC-hub interface illuminated by GFP-E-cadherin (DEFL, green) expressed in the germline (nos-gal4>UAS-DEFL). (**D**) Schematic describing the definition of centrosome orientation and Baz-centrosome docking. DOI: 10.7554/eLife.04960.003

Because we previously found that Par-1 is a critical component of the COC (*Yuan et al., 2012*), we examined the role of Baz in the centrosome orientation and/or COC. Baz has been reported to localize at the hub-GSC interface along with E-cadherin following overexpression in the germline (nos > Baz-GFP) (*Leatherman and Dinardo, 2010*), which was confirmed by using independent UAS-Baz-YFP construct (see below). However, closer inspection using antibody staining and Flytrap Baz-GFP that expresses endogenous levels of Baz [CC01941 (*Kelso et al., 2004*; *Buszczak et al., 2007*)] revealed that Baz forms foci at the hub-GSC interface (referred to as the 'Baz patch' hereafter), instead of entirely colocalizing with E-cadherin (*Figure 1B,C*). The Baz patch is a small structure, with a size of approximately 1.5 μm, and this patch is considerably smaller than the GSC-hub interface that is marked by E-cadherin (4–6 μm) (*Figure 1C*). We noticed that the Baz patch was often closely associated with the apical centrosome (68.8 ± 2.2% of total GSCs; *Figure 1C*, arrowheads). We termed this close association of the Baz patch and the centrosome 'Baz-centrosome docking'. Baz-centrosome docking is a more specific criteria compared to centrosome orientation: ~90% of total GSCs had 'oriented' centrosomes, a category that can be further subdivided into GSCs with 'oriented, but not docked' centrosomes (~20%) and those with 'oriented and docked' centrosomes (~70%) (*Figure 1D*). The remaining ~10% of total GSCs had misoriented centrosomes as reported previously (*Figure 1D*) (*Cheng et al., 2008*; *Roth et al., 2012*; *Yuan et al., 2012*).

## Baz-centrosome docking is cell cycle dependent and peaks during the late G2 phase

Using a combination of multiple cell cycle markers, we found that Baz-centrosome docking is cell cycle dependent, reaching a peak of ~80% during late G2 phase. The GSC cell cycle was judged by the

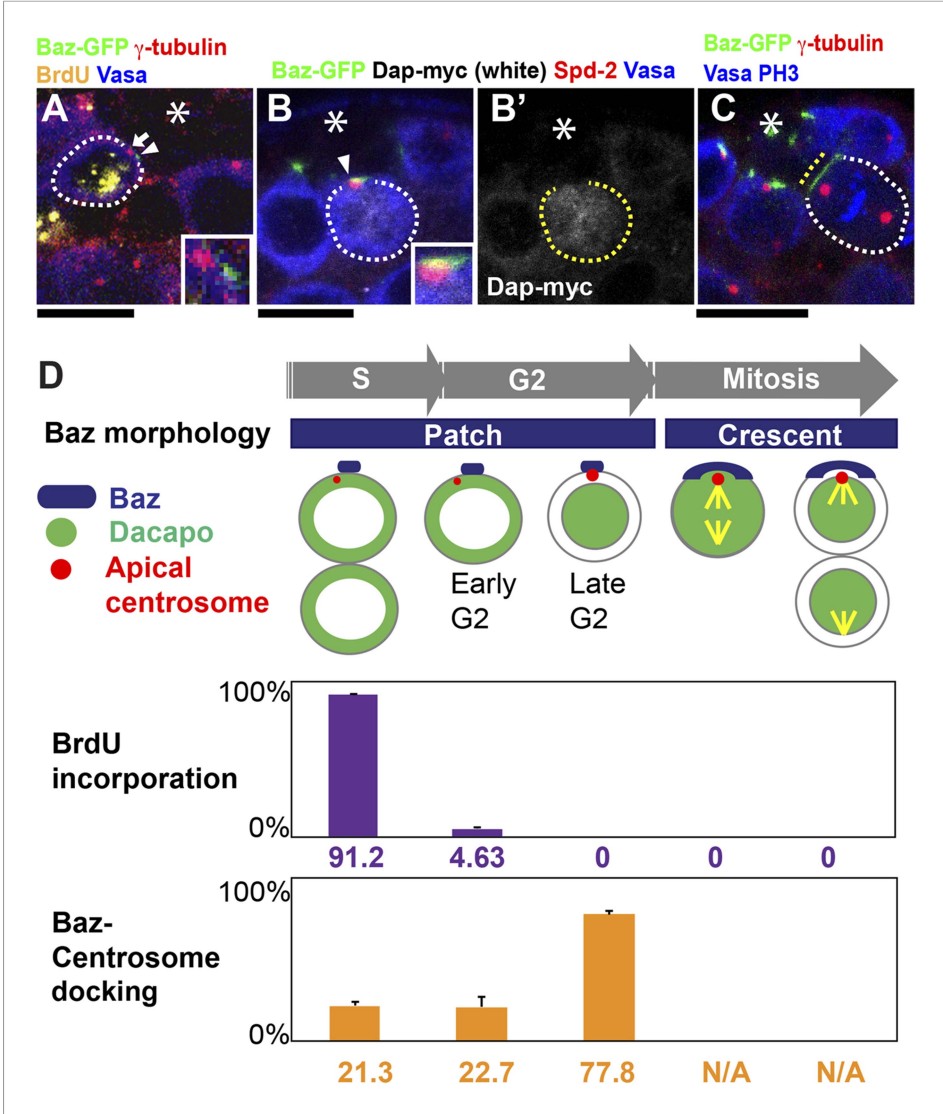

**Figure 2**. Baz-centrosome docking is cell cycle-dependent. (**A**) A representative image of an undocked centrosome in S phase. The arrow indicates the centrosome, and the arrowhead indicates the Baz patch. The inset shows a magnified view. (**B**) A representative image of a late G2 GSC with nuclear Dap (white). The arrowhead indicates the centrosome docked to the Baz patch. Spd-2 staining indicates the centrosome (red). (**B'**) Dap-myc only. (**C**) A representative image of a mitotic GSC. The yellow dotted line indicates a Baz crescent along the hub-GSC interface. At this point, Baz-centrosome docking cannot be assessed because Baz does not localize as foci ('N/A' in panel **D**). (**D**) The frequency of Baz-centrosome docking during the cell cycle.

following criteria (*Figure 2*). (1) A short pulse of ex vivo BrdU incorporation was used to detect cells in S phase. GSCs that were positive for BrdU were always connected with their differentiating daughters (GBs), which were also positive for BrdU in synchrony. This finding suggests that GSCs (and GBs) enter S phase prior to cytokinesis and that the G1 phase is extremely short. (2) As cells progress into G2 phase, GSCs started to accumulate Dap, a Cip/Kip family CDK inhibitor, in the nucleus, as reported previously (*Meyer et al., 2002*). Thus, late G2 GSCs were detected as nuclear Dap-positive cells. (3) Prior to nuclear Dap accumulation, GSCs that completed cytokinesis, negative for BrdU and had not accumulated nuclear Dap were judged as early G2 phase.

Using these criteria, we correlated the cell cycle stages with the Baz-centrosome docking status. During S phase, the frequency of Baz-centrosome docking was low (~20%, *Figure 2A,D*). GSCs in early G2 phase maintained low Baz-centrosome docking (*Figure 2D*); however, once GSCs reached

late G2 phase, Baz-centrosome docking was elevated dramatically, reaching approximately 80% (*Figure 2B,D*). As GSCs entered mitosis, Baz was broadly distributed to the hub-GSC interface instead of being confined as a small patch (*Figure 2C*). This distribution resembled the Baz crescent that was observed in *Drosophila* neuroblasts (*Schober et al., 1999*; *Wodarz et al., 1999*). At this point in the cell cycle, Baz-centrosome docking could no longer be defined due to diffuse Baz localization (*Figure 2D*, N/A). Together, these data demonstrate that Baz-centrosome docking is a cell cycle-dependent event that occurs just before mitotic entry.

## Baz plays a critical role in the centrosome orientation and the COC

The tight association (docking) of the Baz patch and the centrosome just before mitotic entry led us to hypothesize that such association may be the physical basis for 'correct centrosome orientation' that allows GSCs to enter mitosis. We speculated that the Baz patch may provide a docking site for the centrosome to correctly orient and that once such docking occurs, GSCs may interpret this docking as 'correct centrosome orientation'.

To address these possibilities, we first examined the potential role of Baz in centrosome orientation and COC function. We used centrosome and spindle misorientation as the major criteria for assessment of the function of centrosome orientation mechanism and the COC. If the centrosome orientation mechanism is defective, a high frequency of centrosome misorientation would result, although this scenario does not necessarily lead to spindle misorientation. If the COC is intact, GSCs with misoriented centrosomes would halt cell cycle progression prior to mitotic entry, resulting in a low frequency of spindle misorientation, even if the centrosomes are highly misoriented (*Figure 1A*). If the COC is also defective, then GSCs with misoriented centrosomes would enter mitosis unchecked, resulting in a high frequency of misoriented spindles.

We first attempted to knock down Baz using two independent RNAi lines (validated in *Figure 3—figure supplement 1*). In control GSCs, centrosomes were oriented in most of the GSCs (up to ~10% centrosome misorientation, *Figure 3A,G*) as reported previously (*Yamashita et al., 2003*, *2007*). On the contrary, RNAi-mediated knockdown of Baz caused a high frequency of centrosome misorientation in interphase GSCs (*Figure 3B,G*). This result suggests that Baz is required for normal centrosome orientation in GSCs. Combined with the observation that the centrosome docks to the Baz patch, these results indicate that the Baz patch provides a physical platform for GSC centrosome association to achieve correct centrosome orientation. Baz RNAi caused a minor but statistically significant increase in the spindle misorientation (*Figure 3C, D, G*), suggesting that Baz is also required for COC at least partially.

To further explore the potential function of Baz, we overexpressed Baz in the germline (nos > Baz-YFP). Upon overexpression, Baz often formed ectopic patches outside the hub-GSC interface (*Figure 3E*, arrow) and often broadly localized to the hub-GSC interface (*Figure 3F*) as reported previously (*Leatherman and Dinardo, 2010*). Ectopic Baz patches were often associated with misoriented centrosomes (*Figure 3E*, inset), suggesting that the Baz patch has the ability to dock to centrosomes ectopically, even outside the hub-GSC interface. Strikingly, Baz overexpression led to a high frequency of spindle misorientation as well as centrosome misorientation (*Figure 3F,G*). These data indicate that the Baz patch can ectopically dock the centrosome, and this docking is sufficient to satisfy the requirement of the COC, allowing GSCs to enter mitosis with misoriented spindles (*Figure 3H*).

The status of COC activity was further assessed by a recently developed assay, in addition to scoring the centrosome and spindle misorientation. Recently, we showed that the treatment of testes with colcemid, a microtubule depolymerizing agent, can serve as a sensitive assay to monitor the COC activity (*Venkei and Yamashita, 2015*). When testes were incubated with colcemid, spermatogonia, the differentiating progeny of GSCs, arrest in mitosis due to the activation of the spindle assembly checkpoint (SAC). On the contrary, GSCs arrest in G2 phase of the cell cycle instead of mitosis due to centrosome misorientation and activation of COC, which operates prior to SAC. In the absence of functional COC (such as in *par-1* mutant), however, colcemid treatment leads to SAC-mediated mitotic arrest by bypassing COC-mediated G2 arrest (*Figure 3I*). Thus, accumulation of mitotic GSCs in the presence of colcemid serves as a sensitive readout of defective COC. By using this method, we assessed the activity of COC in control, Baz RNAi and Baz-overexpressing GSCs (*Figure 3I*). Indeed, the results confirmed the model obtained by the scoring of centrosome/spindle orientation: (1) Baz RNAi GSCs maintain relatively intact COC activity, and (2)

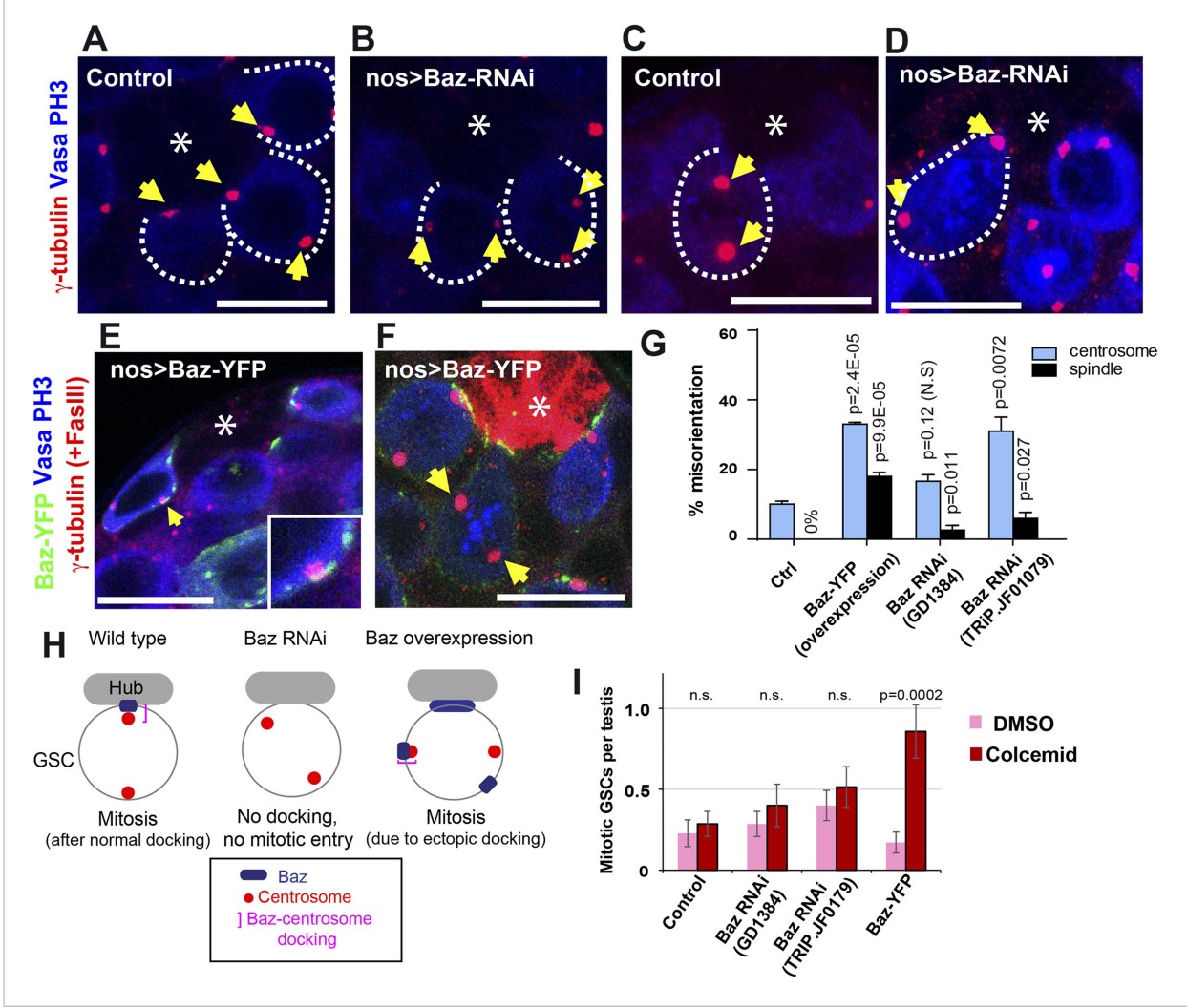

**Figure 3.** Baz is required for centrosome orientation. (**A, B**) Control (**A**) and Baz RNAi (**B**) testes showing GSCs in interphase. Arrows indicate centrosomes. GSCs are indicated by broken lines. The hub is denoted by an asterisk. Bar: 10 μm. (**C, D**) Control (**C**) and Baz RNAi (**D**) GSCs in mitosis. Arrows indicate spindle poles. (**E**) Overexpressed Baz-YFP ectopically localizes to the lateral cortex of GSCs. Arrows indicate the ectopic patch docking to the centrosome. (**F**) Mitotic GSCs with misoriented spindles upon overexpression of Baz-YFP. (**G**) Frequencies of centrosome (% of total GSC) and spindle (% of mitotic GSC) misorientation upon Baz RNAi or Baz-YFP overexpression. N > 300 GSCs were scored for centrosome orientation, and N > 30 mitotic GSCs were scored for spindle orientation. (**H**) A model for Baz-centrosome docking and mitotic entry in control, Baz RNAi, and Baz overexpression. (**I**) Mitotic index of GSCs after incubation with or without colcemid for 4.5 hr in indicated genotypes. Increased mitotic index in the presence of colcemid indicates defective COC. p value indicates the statistical significance in an increase in mitotic index in the presence of colcemid.

The following figure supplement is available for figure 3:

**Figure supplement 1.** GFP fluorescent quantification of Baz patch (Baz-GFP Flytrap) upon knockdown of Baz (GD1384 and JF01079).

Baz-YFP overexpression allows GSCs to enter mitosis, presumably through ectopic Baz-centrosome docking, which inactivates COC activity.

In summary, the data presented here point to a model in which the Baz patch is a platform for centrosome anchoring, and the Baz-centrosome association (docking) is the cellular process that passes the COC to permit mitotic entry (*Figure 3H*). Considering the (partial) requirement of Baz in COC, we speculate that Baz may play dual roles in COC, depending on the centrosome-docking status: it might contribute to activation of COC when undocked with centrosomes, whereas it might inactivate COC when docked with centrosomes.

## Baz functions downstream of E-cadherin

Similar to Baz overexpression, overexpression of a dominant-negative E-cadherin (dCR4h) leads to a high frequency of misoriented spindles (*Inaba et al., 2010*). dCR4h lacks the extracellular domain and thus cannot engage in the homotypic interaction (*Oda and Tsukita, 1999*). As a result, overexpressed dCR4h localizes to the entire cortex of GSCs instead of being limited to the hub-GSC interface (*Inaba et al., 2010*). Therefore, we speculated that the cytoplasmic domain of E-cadherin may participate in anchoring of the centrosomes, and ectopic anchoring between dCR4h and the centrosome may satisfy the conditions required to inactivate the COC to allow mitotic progression.

Because Baz/Par-3 is recruited to adherens junctions (*Le Borgne et al., 2002*), we speculated that Baz might function downstream of E-cadherin to anchor the centrosomes. To test this possibility, we first investigated the effect of dCR4h expression on Baz localization. GSC clones that express dCR4h were induced by heat-shock treatment (hs-FLP, nos > stop > gal4, UAS-dCR4h-GFP, UAS-GFP). In control GFP-negative GSCs, the Baz patch was observed as described above (*Figure 4A*, arrow). On the contrary, in GSCs that expressed dCR4h, Baz broadly localized along the hub-GSC interface, as opposed to concentrated localization as a patch (*Figure 4A*, broken lines), and the frequency of Baz patch-positive GSCs was significantly reduced (*Figure 4B*). Furthermore, we observed that overexpression of dCR4h often formed ectopic Baz patches away from the hub-GSC interface, and such a Baz patch was associated with the centrosomes (approximately 10% of GSCs expressing dCR4h, *Figure 4C*). In addition, Baz patch was undetectable in most of GSC clones that are homozygous for E-cadherin loss of function mutation (*shg*[10469]) confirming that Baz localization depends on E-cadherin (*Figure 4D,E*). These data indicate that the cytoplasmic tail of E-cadherin indeed recruits Baz, which in turn anchors the centrosome.

If this model is correct, the ability of dCR4h to misorient the spindles may rely on the presence of Baz. To test this possibility, we examined the effect of Baz knockdown on dCR4h-mediated centrosome/spindle misorientation. Overexpression of dCR4h alone led to a high frequency of spindle misorientation, as reported previously (*Figure 4F, H*) (*Inaba et al., 2010*); however, co-expression of Baz RNAi with dCR4h significantly suppressed spindle misorientation (*Figure 4G, H*). These results clearly demonstrate that the spindle misorientation caused by dCR4h overexpression is due to ectopic recruitment of Baz, which in turn anchors the centrosomes.

By using colcemid treatment, we further assessed COC status under these conditions. Expression of dCR4h resulted in mild abrogation of COC (*Figure 4I*), as predicted by high frequency of spindle misoreintation upon expression of dCR4h (*Figure 4H*). Interestingly, despite rescue of spindle misorientation by Baz RNAi, COC defect was not rescued under these conditions (*Figure 4I*). We speculate that Baz RNAi reduces the level of ectopic Baz due to dCR4h overexpression, leading to reduced level of spindle misorientation, without rescuing COC defect.

Taken together, these results show that Baz functions downstream of E-cadherin to anchor the centrosome, and that E-cadherin-Baz-centrosome interaction is the critical aspect in satisfying the COC (*Figure 4J*).

## Par-1-dependent phosphorylation of Baz is required for the GSC spindle orientation

The above data suggest that Baz is a critical component of the GSC centrosome orientation and that Baz-centrosome docking is the physical basis that is monitored by the COC. Our previous study demonstrated that Par-1, a physiological kinase of Baz, is a critical component of the COC (*Yuan et al., 2012*). The mechanism by which Par-1 mediates sensing of the correct centrosome position is unknown. Therefore, we set out to examine whether phosphorylation of Baz by Par-1 is important for COC function.

Two conserved serine residues of Baz protein, serine 151 (S151) and serine 1085 (S1085), are known to be phosphorylated by Par-1 (*Krahn et al., 2009*). To begin to address the relationship between Baz and Par-1 in the COC, we first examined the phosphorylation status of Baz using phosphorylation-specific antibodies against phospho-S151 (pS151) and phospho-S1085 (pS1085) (*Krahn et al., 2009*). We detected pS151 (*Figure 5A*), but not pS1085, at the Baz patch; thus, we focused only on pS151 function in subsequent experiments. Interestingly, phosphorylation of the Baz patch is dependent on the cell cycle and/or centrosome orientation status. Baz phosphorylation was considerably weaker in GSCs in early G2 phase with undocked but oriented centrosomes (*Figure 5B,D*)

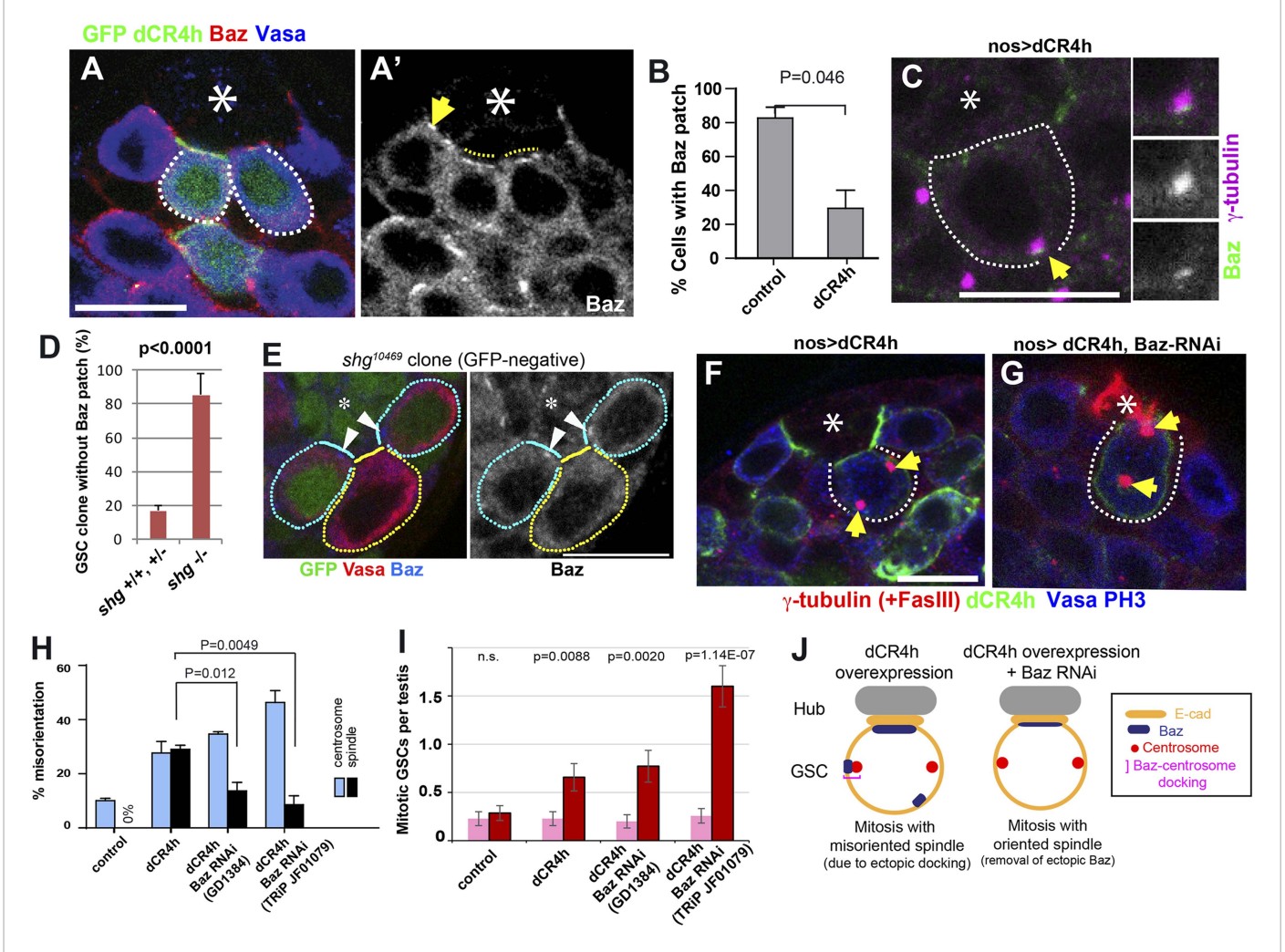

**Figure 4**. Baz functions downstream of E-cadherin during centrosome orientation. (**A**) An apical tip of the testis containing clones (GFP+) that express a dominant-negative form of E-cadherin (dCR4h). The arrow indicates normal Baz patch in a control GSC that does not express dCR4h. Broken lines indicate diffused Baz localization upon expression of dCR4h. (**A'**) anti-Baz only. The hub is denoted by an asterisk. Bar: 10 μm. (**B**) Frequency of GSCs with the Baz patch in control vs dCR4h-expressing GSCs. N > 100 GSCs were scored. (**C**) An example of ectopic Baz patch away from the hub-GSC interface (arrow) that docks the centrosome upon expression of dCR4h. (**D**) Frequency of Baz patch in control vs shg10469 (E-cadherin loss of function allele) GSC clones. (**E**) An example of GFP-, shg10469 clone without Baz patch (yellow line). Control (GFP+) GSCs with Baz patch (arrowheads) are juxtaposed (cyan lines). (**F**) An example of a misoriented spindle upon expression of dCR4h. Arrows indicate spindle poles. (**G**) An example of oriented spindles in GSCs that express both dCR4h and Baz RNAi. (**H**) Frequencies of centrosome (% of total GSC) and spindle (% of mitotic GSC) misorientation upon expression of dCR4h in the presence or absence of Baz RNAi. N > 300 GSCs were scored for centrosome orientation, and N > 30 mitotic GSCs were scored for spindle orientation. (**I**) Mitotic index of GSCs after incubation with or without colcemid for 4.5 hr indicate genotypes. Increased mitotic index in the presence of colcemid indicates defective COC. p value indicates the statistical significance in an increase in mitotic index in the presence of colcemid. (**J**) A model for Baz-centrosome docking and mitotic entry in dCR4h-expressing GSCs with or without Baz RNAi.

compared to that in GSCs in late G2 phase when the centrosome is docked to the Baz patch (*Figure 5A,D*). Once cells entered mitosis, Baz phosphorylation became undetectable (*Figure 5C*), as the Baz patch diffused (*Figure 2C*). Using Par-1 RNAi, which was previously validated in GSCs (*Yuan et al., 2012*), we found that the pS151 signal intensity was significantly reduced in Par-1 RNAi GSCs (*Figure 5E*), suggesting that Par-1 is required for phosphorylation of Baz-S151, as has been shown in other cell types (*Benton and St Johnston, 2003*; *Krahn et al., 2009*).

To address whether this Par-1-dependent phosphorylation of Baz is important in the COC function, we examined the effect of overexpression of a non-phosphorylatable form or

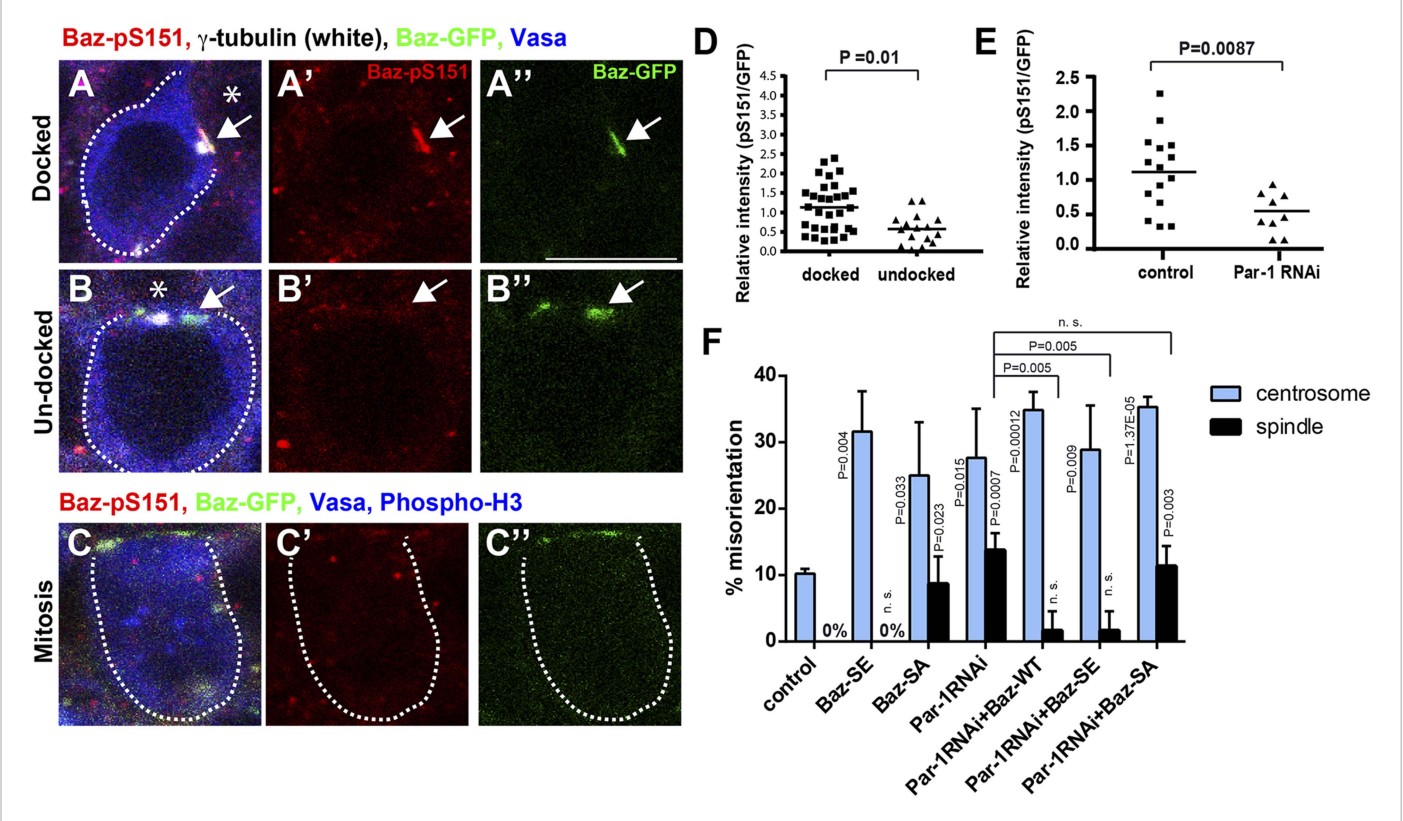

**Figure 5**. Par-1-dependent Baz-S151 phosphorylation is required for the centrosome orientation checkpoint. (**A–C**) Phosphorylation of Baz-S151 was monitored by a phospho-S151-specific antibody during cell cycle. The Baz-patch is indicated by the arrow. Red indicates Baz-pS151, green indicates Baz-GFP, and blue indicates Vasa. Bar: 10 µm. The hub is denoted with an asterisk. (**D**) The quantification of Baz-S151 phosphorylation levels. The signal was normalized by Baz-GFP (pixel intensity of pS151 staining was divided by the pixel intensity of Baz-GFP). The background (cytoplasm signal in the same cell) was subtracted from both pS151 and Baz-GFP prior to calculation. N > 10 GSCs were scored. (**E**) The quantification of Baz-S151 phosphorylation level in control vs Par-1 RNAi GSCs. N > 10 GSCs was scored. (**F**) Frequencies of centrosome (% of total GSC) and spindle (% of mitotic GSC) misorientation in control vs Par-1 RNAi GSCs with or without expression of wild type Baz, Baz-SA or Baz-SE. N > 300 GSCs were scored for centrosome orientation, and N > 30 mitotic GSCs were scored for spindle orientation.

a phosphomimetic form of Baz. Upon expression of the nonphosphorylatable form of Baz (Baz-SA, Baz-S151A S1085A), GSCs showed a high frequency of centrosome and spindle misorientation (*Figure 5F*). These results clearly demonstrate that the phosphorylation of Baz by Par-1 is critical for COC function. Furthermore, we found that overexpression of wild-type Baz as well as the phosphomimetic form of Baz (Baz-SE, Baz-S151E) suppressed spindle misorientation caused by Par-1 RNAi (*Figure 5F*). In contrast, overexpression of the non-phosphorylatable form of Baz (Baz-SA) did not rescue spindle misorientation due to Par-1 RNAi (*Figure 5F*). These results clearly demonstrate that Par-1 executes its function in the COC mainly through phosphorylation of Baz. Notably, Baz-SE did not rescue the centrosome misorientation phenotype, and overexpression of Baz-SE in a wild-type background resulted in a high frequency of centrosome misorientation, despite near complete suppression of spindle misorientation caused by Par-1 RNAi. These observations suggest that although phosphorylation of Baz is sufficient for COC function, dephosphorylation is also important for anchoring the centrosomes. The cell cycle-dependent phosphorylation cycle of Baz (*Figure 5A–D*) is also consistent with the idea that the phosphorylation and dephosphorylation cycle of Baz may be important for anchoring the centrosomes and monitoring centrosome orientation. Taken together, these results reveal the critical function of Par-1-mediated Baz phosphorylation in the function of the COC.

## Impact of COC defects on asymmetric outcome of GSC divisions

To further assess the outcome of COC defect in these mutant conditions described above, we scored the frequency of symmetric GSC divisions. After inducing GSC clones at a low frequency (i.e., dominantly single GSC clone/testis), symmetric outcome can be assessed by scoring the frequency of 'doublet' clones (i.e. two GSC clones are juxtaposed each other) (*Figure 6A–C*) (*Salzmann et al., 2013*). Wild-type GSC clones showed a basal level of doublet frequency (~10%)

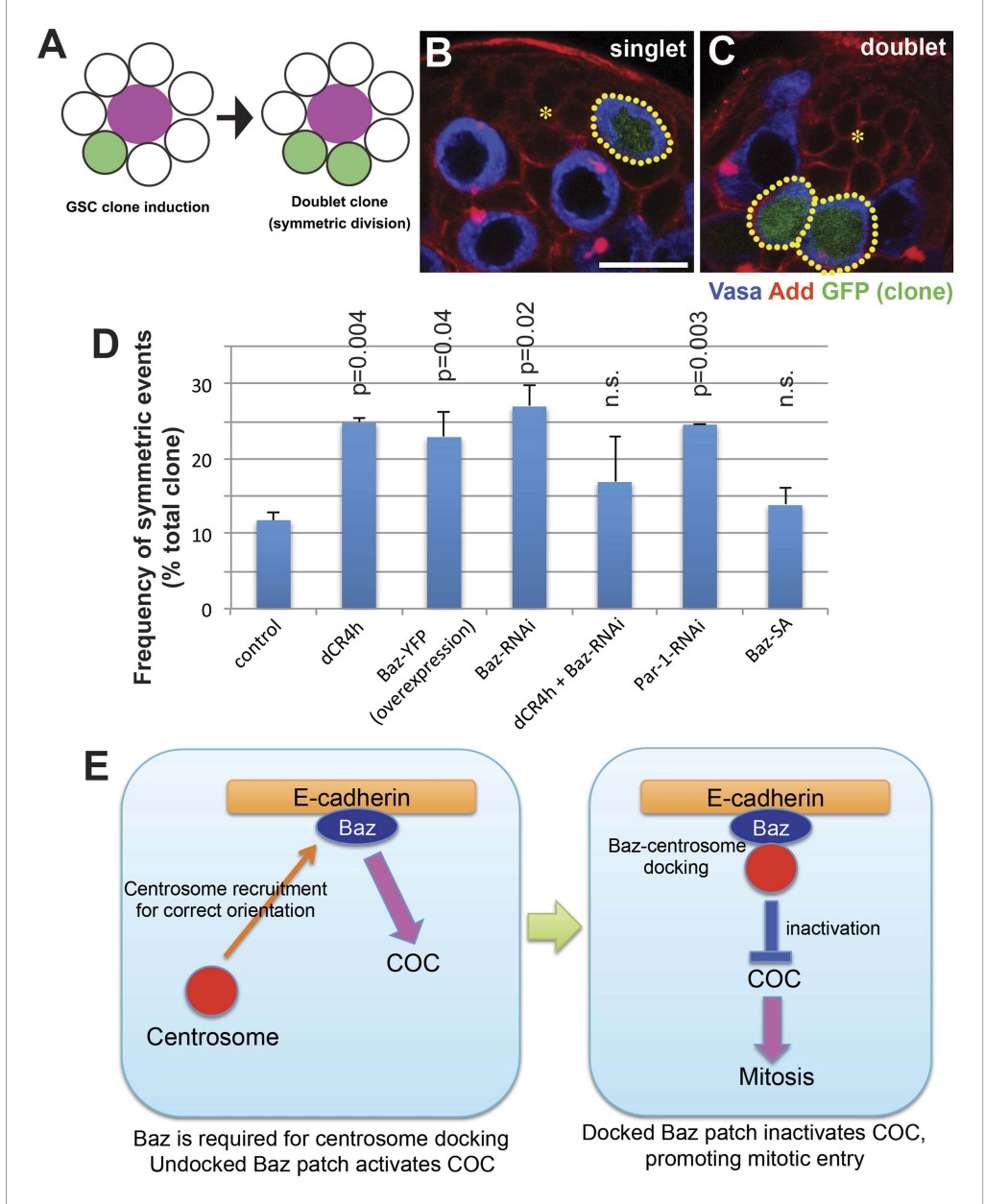

**Figure 6**. COC is required to prevent symmetric GSC divisions. (**A**) An assay system to examine symmetric GSC divisions. GFP+ clone is induced at a low frequency using hs-FLP, nos > stop > gal4, UAS-GFP by a 20-min heatshock. GFP clones were examined 24 hr post heatshock. When such GSCs undergo symmetric stem cell division, it will generate doublet clones (two GFP+ GSCs are juxtaposed each other). (**B**, **C**) Representative images of singlet (**B**) and doublet (**C**) GSC clones. Hub is indicated by the asterisk. Clones are indicated by dotted lines. Bar: 10 μm. (**D**) Frequency of doublets after 24 hr post heatshock. JF01079 line was used for Baz RNAi. (**E**) Model of Baz function in COC (see text for detail).

(*Figure 6D*). This is due to expected low frequency events: (1) two juxtaposing GSCs become clones independently, and (2) 'crawling back' of GBs to the niche causes symmetric outcome of the division as described previously (*Sheng and Matunis, 2011*). dCR4h clones showed increased doublet frequency due to symmetric divisions as described previously (*Salzmann et al., 2013*). GSC clones of Baz overexpression, Baz RNAi or Par-1 RNAi showed significant increase in the frequency of doublet clones (*Figure 6D*). In contrast, GSC clones of Baz-SA as well as dCR4h with Baz RNAi did not increase the frequency of doublet clones. We did not examine Baz-SE, since Baz-SE yielded extremely low frequency of GSC clone induction for unknown reasons. These results demonstrate a correlation between COC defect/spindle misorientation and symmetric GSC division. However, the extent of spindle misorientation and the doublet clone frequency did not perfectly correlate, suggesting that there may be additional mechanisms to contribute to symmetric outcome: for example, spindle misorientation may not necessarily lead to symmetric GSC divisions, if spindle orientation is corrected after entering mitosis or the GSC clones are defective in niche adhesion in addition to spindle orientation.

## Discussion

Although intensive investigations have revealed the mechanisms of cell polarity and asymmetrical cell division along the polarity axis, much less is known about how cells ensure the correct temporary order of cell polarization and cell division. Precocious cell division before establishment of correct polarity would lead to a deleterious outcome, such as a failure in cell fate determination; however, the presence of checkpoint mechanisms to ensure asymmetric division has not been thoroughly investigated. In *Drosophila* neuroblasts, which divide asymmetrically by stereotypically oriented spindles, a phenomenon called 'telophase rescue' has been reported: many mutants that compromise correct spindle orientation in neuroblasts eventually divide asymmetrically (*Lu et al., 1998*; *Schober et al., 1999*; *Wodarz et al., 1999*; *Peng et al., 2000*). In 'telophase rescue', asymmetric outcome of the division is restored by correcting the localization of basal polarity proteins (*Schober et al., 1999*; *Peng et al., 2000*). Such correction might indicate the presence of an orientation/polarity checkpoint, although the mechanistic basis remains unknown. Thus, the COC may serve as a model system to study a new class of checkpoints that specialize in monitoring division orientation in multicellular organisms.

In the present study, we showed that Baz is a critical player in centrosome orientation and its checkpoint in *Drosophila* male GSCs. Baz forms a subcellular structure (Baz patch) at the hub-GSC interface, which anchors the apical centrosome prior to mitotic entry. Our data indicate that Baz-centrosome docking is the cellular event that is recognized by the COC as correct centrosome orientation. The data presented in this study point to the following working model (*Figure 6E*): (1) Baz patch is formed at the hub-GSC interphase in an E-cadherin-dependent manner. (2) Baz patch functions to recruit the centrosome. In the absence of Baz, centrosome is highly misoriented. (3) Baz is (partially) required for the COC activity. Baz patch that is not docked to the centrosome might contribute to the activation of COC to prevent mitotic entry. (4) Once Baz patch is docked to the centrosome, this docking is interpreted as 'correct centrosome orientation', leading to inactivation of COC and thus mitotic entry. This model indicates that Baz plays dual roles in COC depending on its centrosome-docking status: in the absence of docking, Baz patch activates COC, whereas centrosome-docked Baz functions to inactivate COC.

Our results show that Par-1-mediated phosphorylation of Baz is critical for spindle orientation, although the mechanistic details of phosphorylated Baz function are yet to be determined. The fact that both phosphomimetic form (Baz-SE) as well as wild-type Baz can rescue spindle misorientation in Par-1 RNAi GSCs suggests that timing of phosphorylation might not be so critical, in spite of observed phosphorylation–dephosphorylation cycle of Baz during normal cell cycle. With the currently available data, it is unclear how temporal regulation of Baz phosphorylation relates to steps of Baz-centrosome docking, mitotic entry, and spindle orientation. Furthermore, it is puzzling that overexpression of Baz causes high frequency of spindle misorientation in wild type, whereas the overexpression of the same construct in Par-1 RNAi background lowers spindle misorientation. Future investigation is required to understand how distinct isoforms of Baz (phosphorylated vs non-phosphorylated) participate in distinct aspects of centrosome/spindle orientation.

In summary, our study reveals a cellular mechanism by which stem cells integrate information about cell polarity to regulate their cell cycle progression. Such a mechanism ultimately functions

to ensure the asymmetric outcome of stem cell division. We speculate that the orientation checkpoint may be present in many other multicellular organisms, and the understanding of the COC in *Drosophila* may provide a conceptual framework for understanding orientation checkpoint mechanisms in general.

## Materials and methods

### Fly husbandry and strains

All fly stocks were raised on standard Bloomington medium at 25°C. The following fly stocks were used: UAS-Baz RNAi (TRiP.JF01079, obtained from the Bloomington Stock Center); UAS-Baz RNAi (GD1384, obtained from the Vienna *Drosophila* Center); Baz-GFP (Flytrap project [*Morin et al., 2001*; *Kelso et al., 2004*; *Buszczak et al., 2007*]); nos-gal4 (*Van Doren et al., 1998*), UAS-Baz-YFP (obtained from Cheng-Yu Lee); UAS-Baz-S151A S1085A (*Benton et al., 2002*), UAS-Par-1 RNAi (a kind gift from Bingwei Lu [*Zhang et al., 2007*]); UAS-dCR4h, UAS-DEFL (a kind gift from Hiroki Oda [*Oda and Tsukita, 1999*]); dap1gm-(myc) (a kind gift from Christian F Lehner [*Meyer et al., 2002*]), FRT42D *shg*$^{10469}$ (*Uemura et al., 1996*) and nos > stop > gal4 (*Salzmann et al., 2013*). For construction of Baz-S151E, a point mutation was introduced at the S151 residue by site-directed mutagenesis using the polymerase chain reaction (PCR), and the mutant was subcloned into pUAST-EGFP-attB. All transgenic flies were generated using PhiC31 integrase-mediated transgenesis systems (*Groth et al., 2004*) by BestGene, Inc.

### Immunofluorescence staining and confocal microscopy

Immunofluorescence staining was performed as described previously (*Cheng et al., 2008*). The following primary antibodies were used: mouse anti-γ-tubulin (1:100; GTU-88, Sigma-Aldrich, St. Louis, MO), mouse anti-Fasciclin III (FasIII; 1:20; developed by C Goodman and obtained from the Developmental Studies Hybridoma Bank [DSHB], Iowa City, IA), rabbit anti-Thr3-phosphorylated histone H3 (1:200; Cell Signaling Technology, Danvers, MA), rabbit anti-Vasa (1:100; Santa Cruz Biotechnology, Santa Cruz, CA), rat anti-Vasa (1:20; developed by AC Spradling and D Williams and obtained from DSHB), mouse anti-c-myc (1:100; clone 9E10, DSHB), rabbit anti-c-myc (1:30; c3956; Sigma-Aldrich, St. Louis, MO), rabbit anti-Spd-2 (*Giansanti et al., 2008*) (a kind gift from Maurizio Gatti, Dipartimento di Biologia e Biotecnologie Università di Roma), guinea pig anti-Baz (1:500; from Cheng-Yu Lee [University of Michigan] and Chris Doe [University of Oregon]), and rabbit anti-Baz-pS151 and Baz-pS1085 (*Krahn et al., 2009*) (a kind gift from Andreas Wodarz [Georg-August-Universitat Gottingen]). Guinea pig anti-Baz (1:10,000) was also generated using the synthetic peptide Ac-VSEPDASKPRKTWLLEDGDHEGGFASQRC-amide (Covance, Denver, PA), which showed the same staining pattern with other anti-Baz antibodies, thus used interchangeably in the experiments reported here. AlexaFluor-conjugated secondary antibodies were used at a dilution of 1:200 (Life Technologies, Carlsbad, CA). Images were taken using a Leica TCS SP5 confocal microscope with a 63× oil immersion objective (NA = 1.4) and processed using Adobe Photoshop software.

### In vitro BrdU labeling

45-min ex vivo BrdU pulse labeling was performed as previously described (*Roth et al., 2012*). BrdU was detected by immunofluorescence staining using rat anti-BrdU antibody (1:50; Abcam, ab6326, Cambridge, MA).

### Data analyses

Statistical analysis was performed using Microsoft Excel 2010 or GraphPad Prism 6 software. Pixel intensity analyses for staining of Baz phospho-specific S151, nuclear Dap, and Baz-GFP were performed using ImageJ software. For centrosome and spindle orientation scoring, >300 GSCs were scored for centrosome misorientation, and >30 mitotic GSCs were scored for spindle misorientation. Centrosome misorientation was indicated when neither of the two centrosomes were closely associated with the hub-GSC interface during interphase. Spindle misorientation was indicated when neither of the two spindle poles were closely associated with the hub-GSC interface during mitosis. Data are shown as means ±standard deviation. The p-value (two-tailed student's *t*-test) is provided for comparison with the control.

## Acknowledgements

We thank Drs Chris Doe, Cheng-Yu Lee, Andreas Wodarz, Daniel St Johnston, Maurizio Gatti, Elizabeth R Gavis, Bingwei Lu, Hiroki Oda, Christian F Lehner, Bing Ye, and Allan C Spradling as well as the Flytrap project, the Bloomington *Drosophila* Stock Center, the Vienna *Drosophila* Stock Center, and the Developmental Studies Hybridoma Bank for reagents. We thank the Yamashita lab members for discussion. This work was supported by NIH R01GM086481 (to YMY) and Howard Hughes Medical Institute. YMY is supported by the John D and Catherine T MacArthur Foundation.

# Additional information

### Competing interests

YMY: Reviewing editor, *eLife*. The other authors declare that no competing interests exist.

### Funding

| Funder | Grant reference | Author |
| --- | --- | --- |
| National Institutes of Health (NIH) | R01GM086481 | Yukiko M Yamashita |
| Howard Hughes Medical Institute | | Yukiko M Yamashita |
| John D. and Catherine T. MacArthur Foundation | | Yukiko M Yamashita |

The funders had no role in study design, data collection and interpretation, or the decision to submit the work for publication.

### Author contributions

MI, YMY, Conception and design, Acquisition of data, Analysis and interpretation of data, Drafting or revising the article; ZGV, Acquisition of data, Analysis and interpretation of data

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
