## [Decision Letter]

Thank you for sending your work entitled “Baz forms a platform for the centrosome orientation during asymmetric stem cell division in the *Drosophila* male germline” for consideration at *eLife*. Your article has been favorably evaluated by Janet Rossant (Senior editor), a Reviewing editor, and three reviewers.

The Reviewing editor and the reviewers discussed their comments before we reached this decision, and the Reviewing editor has assembled the following comments to help you prepare a revised submission.

Major issues:

While the reviewers agree that the data present new mechanistic insight about centrosome orientation, the authors do not explore the relevance of the described cellular phenotypes to the self-renewal biology of the stem cell system. As a result, the ultimate consequences of the various phenotypes on stem cell self-renewal are unclear. The authors should generate mosaic clones and assay over time for clone size, expression of differentiation markers, and the frequency of stem cell loss.

The mere fact that mis-oriented spindles are seen doesn't really tell us whether or not a checkpoint is activated. In Figure 3, do the cells spend a longer time before entering mitosis when their spindles are mis-oriented in the Baz over-expression line? The prediction of the model is that the time prior to mitosis would be the same even though spindles are now mis-oriented. It will also be valuable to determine the number of stem cells in a Baz-RNAi background.

In the Cadherin experiments, when talking about “fewer spindles”, please be really clear about whether you mean a small fraction of the spindles that are seen, or a smaller total number of spindles. One expects fewer spindles, but the fraction of spindles that were mis-oriented should be the same as when Baz is present. Is this correct? Also in Figure 4, the idea that Cad recruits baz is based solely on overexpression of a dominant negative construct. Loss of function studies should be included. Numerous Ecad alleles of varying strengths are available to facilitate this analysis. Furthermore, in Figure 4, data on mitotic entry is lacking. The prediction is that mitotic index is higher in cadherin DN than cadherin DN + baz RNAi.

The fact that Baz phosphomimetic mutant suppresses the spindle orientation defect of Par-1 RNAi indicates that the phosphorylation must not play a role in deciding when the checkpoint is satisfied, why then is the orientation defect in Par-1 RNAi suppressed by the phosphorylation? Also, an important control is to show that overexpression of wild type Baz does not cause this suppression. In any case, the phosphomimetic and phosphodead experiments are done using overexpression experiments. As overexpression of wild-type Baz causes centrosome and spindle defects, these results are not easily interpretable. Since the alternative, of making transformants with wild-type expression is time consuming, the authors should address this issue as a caveat and provide some explanation as to why these data still remain viable. In the absence of such data, perhaps the authors will want to soften their description of this phenomenon as a “checkpoint”.

---

## [Author Response]

*While the reviewers agree that the data present new mechanistic insight about centrosome orientation, the authors do not explore the relevance of the described cellular phenotypes to the self-renewal biology of the stem cell system. As a result, the ultimate consequences of the various phenotypes on stem cell self-renewal are unclear. The authors should generate mosaic clones and assay over time for clone size, expression of differentiation markers, and the frequency of stem cell loss*.

We have generated GSC clones of various genotypes examined in this study to show that defective COC/spindle orientation results in a higher frequency of symmetric stem cell division (added as Figure 6). We believe that the assay of symmetric stem cell division is the best readout to examine the direct effect of spindle orientation defect. Other parameters, such as stem cell loss and differentiation can be secondary effects or reflecting other functions of the gene (for example, E-cadherin is involved in GSC-hub adhesion).

*The mere fact that mis-oriented spindles are seen doesn't really tell us whether or not a checkpoint is activated. In*
Figure 3*, do the cells spend a longer time before entering mitosis when their spindles are mis-oriented in the Baz over-expression line? The prediction of the model is that the time prior to mitosis would be the same even though spindles are now mis-oriented. It will also be valuable to determine the number of stem cells in a Baz-RNAi background*.

We thank the reviewers for this critically important comment. The key to the concept of the checkpoint is how cells spend time (correcting, waiting) prior to commitment to the next cell cycle stage (i.e. mitotic entry in this case). Although measuring the cell cycle time and spindle orientation in GSCs for genotypes of interest would be the most direct way to test the hypothesis (as suggested by the reviewers), this is not practical for the following reasons: thus, we adapted a new assay to assess the COC activity, which provided significant new insights into COC mechanism, as summarized below.

Reasons that live observation is not practical in addressing COC activity: Cell cycle of GSCs is quite long (calculated to be 12-16 hours as shown in Yadlapalli et al., 2011, JCS), and one would have to observe two successive mitoses of a single GSC to measure the cell cycle time confidently. However, by the time of second mitosis (which can easily go beyond 20-30 hours after starting the ex vivo culture), the condition of the culture would not be an ideal one, leaving a concern on how valid the measured cell cycle time might be.

An alternative approach to assess the activity of COC: We recently reported a more sensitive assay to address COC activity (Venkei and Yamashita, Development 2015). We showed that upon addition of colcemid, whereas spermatogonia (differentiating progeny of GSCs) arrest in mitosis due to spindle assembly checkpoint, much like any other cell types reported to date (including mammalian culture cells etc.), GSCs instead arrest in G2 phase due to activation of COC (microtubule depolymerization leads to centrosome misorientation, which activates COC). We have further shown that, in GSCs defective in COC (such as par-1 mutant), GSCs would enter and arrest in mitosis (now due to the spindle assembly checkpoint, which is independent of COC). Thus, accumulation of mitotic GSCs in the presence of colcemid can serve as a very sensitive assay to detect defective COC.

The results of colcemid assay to assess COC activity in the genotypes dealt with in this manuscript: By using this new assay, we confirmed that 1) Baz is required for centrosome orientation, and plays a relatively minor role in COC, and 2) Baz-overexpression, as well as dCR4h overexpression, leads to defective COC as suggested by high frequency of spindle misorientation. These results thus strengthened our original conclusion.

*In the Cadherin experiments, when talking about* “*fewer spindles*”*, please be really clear about whether you mean a small fraction of the spindles that are seen, or a smaller total number of spindles. One expects fewer spindles, but the fraction of spindles that were mis-oriented should be the same as when Baz is present. Is this correct*?

(This part was further clarified by the editors:)

*In the experiments testing dCR4h expression to spread out the e-cadherin, we are not sure we get the logic of the conclusion of the Baz knockdown:* “*As a result, the COC remains active due to a lack of Baz-centrosome docking, resulting in prevention of the GSCs from entering mitosis and a lower frequency of misoriented spindles.*” *I think what is confusing us is that if Baz is necessary for the checkpoint to be satisfied, then the checkpoint should never be satisfied whether or not the spindle is correctly oriented, and so what we would have expected is fewer spindles, period, but the fraction of spindles that were mis-oriented should be the same as when Baz is present. Perhaps this is a terminology issue, when the authors talk about* “*fewer spindles*”*, they need to be really clear about whether they mean a small fraction of the spindles that are seen, or a smaller total number of spindles*.

We really appreciate this insightful comment. In the original manuscript, we interpreted the low frequency of spindle misorientation in GSCs of “Baz-RNAi + dCR4h overexpression” as indicating that the lack of Baz prevents COC clearance, explaining low frequency of misoriented spindles. However, contrary to this interpretation, we found that Baz-RNAi + dCR4h GSCs arrest in mitosis (instead of G2) upon colcemid treatment, suggesting that these GSCs are defective in COC or COC is falsely satisfied in this condition. This result indeed revealed that the reviewers’ insight was correct, and that they had sensed something that was not entirely logical in our original manuscript. We are very thankful that the reviewers prevented us from publishing the wrong interpretation we had before.

Now we have reached a better working model. Specifically, we revised our idea as following:

In our original manuscript, we speculated that Baz-centrosome docking is sufficient to induce mitotic entry (i.e. COC inactivation), but that Baz is not required for COC. However, our new data suggest Baz’s dual roles in centrosome orientation and COC (model: Figure 6), possibly depending on centrosome docking status: when Baz is docked with the centrosome, it signals to inactivate COC as proposed previously. In addition, we now speculate that Baz is (at least in part) required to activate COC when Baz patch is not docked with centrosome. The detailed discussion on this point is added to the revised text.

*Also in*
Figure 4*, the idea that Cad recruits baz is based solely on overexpression of a dominant negative construct. Loss of function studies should be included. Numerous Ecad alleles of varying strengths are available to facilitate this analysis*.

We have added the scoring of Baz patch formation in GSC clones that are null for E-cadherin (FLP-FRT mediated mosaic clone using *shg*^*10469*^ (E-cadherin mutant)). This experiment confirmed that E-cadherin is required for Baz patch formation. The result is added to Figure 4.

*Furthermore, in*
Figure 4*, data on mitotic entry is lacking. The prediction is that mitotic index is higher in cadherin DN than cadherin DN + baz RNAi*.

As described above, the reviewers’ suggestion led us to conduct additional experiments, which are now incorporated in the revised manuscript as a new model. Again, we are very grateful to the reviewers for helping us to reach a better understanding of the mechanism.

*The fact that Baz phosphomimetic mutant suppresses the spindle orientation defect of Par-1 RNAi indicates that the phosphorylation must not play a role in deciding when the checkpoint is satisfied, why then is the orientation defect in Par-1 RNAi suppressed by the phosphorylation? Also, an important control is to show that overexpression of wild type Baz does not cause this suppression. In any case, the phosphomimetic and phosphodead experiments are done using overexpression experiments. As overexpression of wild-type Baz causes centrosome and spindle defects, these results are not easily interpretable. Since the alternative, of making transformants with wild-type expression is time consuming, the authors should address this issue as a caveat and provide some explanation as to why these data still remain viable. In the absence of such data, perhaps the authors will want to soften their description of this phenomenon as a* “*checkpoint*”.

We thank the reviewers for this suggestion. We examined the effect of wild type Baz, as well as non-phosphorylatable form of Baz (Baz-SA) in the Par-1 RNAi background. This showed that wild type Baz can also rescue the spindle misorientation phenotype of Par-1 RNAi, whereas Baz-SA does not. These data indicate that: 1) Baz phosphorylation is critical for spindle orientation, 2) but (as reviewer suspected) the timing of phosphorylation might not play a major role in COC. We added these discussions in the main text, and also softened the description of checkpoint as suggested by the reviewers.